# Short-Term Comparison of Switching to Brolucizumab or Faricimab from Aflibercept in Neovascular AMD Patients

**DOI:** 10.3390/medicina60071170

**Published:** 2024-07-19

**Authors:** Akiko Kin, Takahiro Mizukami, Satoru Ueno, Soichiro Mishima, Yoshikazu Shimomura

**Affiliations:** 1Department of Ophthalmology, Fuchu Hospital, Izumi 594-0076, Osaka, Japan; a_kin@seichokai.or.jp (A.K.); s_mishima@seichokai.or.jp (S.M.); y_shimomura@seichokai.or.jp (Y.S.); 2Department of Ophthalmology, PL Hospital, Tondabashi 584-8585, Osaka, Japan; 181226@med.kindai.ac.jp

**Keywords:** age-related macular degeneration, brolucizumab, faricimab, aflibercept, intravitreal injection, switching, vascular endothelial growth factor

## Abstract

*Background and Objectives*: In this study, our objective was to assess and compare the changes in visual and structural outcomes among patients with neovascular age-related macular degeneration (nAMD) who were switched from intravitreal aflibercept (IVA) to either intravitreal brolucizumab (IVBr) or intravitreal faricimab (IVF) injections in a clinical setting. *Materials and Methods*: This observational clinical study included 20 eyes of 20 patients switched to brolucizumab and 15 eyes of 14 patients switched to faricimab from aflibercept in eyes with nAMD. We measured the structural outcome (central macular thickness (CMT)) and the visual outcome (best-corrected visual acuity (BCVA); logMAR) as follows: just before the most recent IVA injection (B0), one month after the most recent IVA injection (B1), just before the first IVBr or IVF injection (A0), one month after (A1) and three months after (A3) the first IVBr or IVF injection. *Results*: BCVA showed significant improvement at A1 (0.25 ± 0.34) and at A3 (0.19 ± 0.24) compared to A0 (0.38 ± 0.35) in the IVBr group (*p* = 0.0156, *p* = 0.0166, respectively). CMT (μm) was significantly thinner at A1 (IVBr: 240.55 ± 51.82, IVF: 234.91 ± 47.29) and at A3 (IVBr: 243.21 ± 76.15, IVF: 250.50 ± 72.61) compared to at A0 (IVBr: 303.55 ± 79.18, IVF: 270.33 ± 77.62) in the IVBr group (A1: *p* = 0.0093, A3: *p* = 0.0026) and in the IVF group (A1: *p* = 0.0161, A3: *p* = 0.0093). There was no significant difference in BCVA and CMT improvement observed between two groups at any time point (*p* > 0.05 for all). *Conclusions*: Switching from aflibercept to either brolucizumab or faricimab has a significant anatomical effect in eyes with nAMD and both treatments appear to be effective short-term treatment options. There is a trend towards greater visual improvements and reductions in CMT with brolucizumab.

## 1. Introduction

Neovascular age-related macular degeneration (nAMD) is the leading cause of permanent blindness among the elderly in industrialized nations [1]. Vascular endothelial growth factor (VEGF) plays a crucial role in the regulation of macular neovascularization (MNV) and vascular permeability [2].

The advent of anti-VEGF therapies has significantly reduced both the prevalence and severity of visual loss in patients with nAMD and has greatly improved their visual outcomes [3]. However, some patients exhibit suboptimal response to anti-VEGF treatment, and a number of them show suboptimal response even after receiving consecutive monthly injections [4,5]. In the VIEW1/VIEW2 trials, 19.7% and 36.6% of patients experienced persistent active exudation after receiving aflibercept (Eylea^®^, Regeneron Pharmaceuticals, Tarrytown, NY, USA) treatment every 4 and 8 weeks for one year, respectively [6]. Approximately 7 years after ranibizumab therapy in the ANCHOR or MARINA trials, one-third of patients had poor outcomes [7]. Over time, 8.9% of patients with nAMD developed resistance, known as tachyphylaxis, after repeated IVA injections, with an annual rate of approximately 3% [8]. Hence, efforts have been made to improve efficacy and extend the duration of the intravitreal injections. Different approaches, such as switching between anti-VEGF agents, adjusting treatment regimens, exploring combination therapies, and employing high-dose treatments, have been investigated in small-scale studies, showing some success over short follow-up periods [9,10,11,12]. Higher-dose therapy tended to improve anatomical outcomes and preserve vision but required frequent injections, approximately one every 5.7 to 6.4 weeks [9]. 

Brolucizumab (Beovu^®^, Novartis International, Basel, Switzerland), a humanized single-chain antibody fragment targeting VEGF, is the smallest among commercially available anti-VEGF agents, allowing for higher-molar-dose administration [13,14]. In two prospective randomized phase 3 trials (HAWK and HARRIER) involving treatment-naïve nAMD patients, brolucizumab showed visual outcomes that were noninferior to aflibercept and exhibited superior anatomical outcomes [14,15]. Faricimab (Vabysmo^®^, Roche/Genentech, Basel, Switzerland) acts through dual inhibition of angiopoietin-2 (Ang-2) and VEGF-A [16]. The TENAYA and LUCERNE clinical trials found that the improvement in best-corrected visual acuity (BCVA) in eyes treated with 6 mg of faricimab was comparable to the improvement seen in eyes treated with 2 mg of aflibercept [17]. 

Many prospective and retrospective, noncomparative studies have been conducted on patients who were switched to a different anti-VEGF agent after not responding to their current treatment regimen [18,19,20]. However, there is currently no consensus on the optimal treatment for eyes with nAMD that have shown suboptimal response to aflibercept. To our knowledge, the outcomes of brolucizumab versus faricimab for eyes with nAMD switched from aflibercept have not been directly compared. Therefore, this study aimed to directly compare the short-term visual and structural outcomes of intravitreal brolucizumab versus faricimab treatment for eyes with nAMD switched from aflibercept in a clinical setting.

## 2. Materials and Methods

### 2.1. Study Population

This study was approved by the Ethics Committee of Fuchu hospital (ID number: 2024009) and adhered to the tenets of the Declaration of Helsinki. Written informed consent was not obtained because the study was retrospective. The need for informed consent was waived by the Ethics Committee of Fuchu eye center. The eye center website provided participants with an opportunity to opt out of the study. This was a retrospective, single-center, observational study. This study included 20 eyes of 20 patients switched to brolucizumab and 15 eyes of 14 patients switched to faricimab from aflibercept for eyes with nAMD. All patients received the treatment from January 2022 until April 2024 and were followed for three months after the switch. 

### 2.2. Intravitreal Injection

All patients received intravitreal injections of aflibercept 2.0 mg/0.05 mL, brolucizumab 6.0 mg/0.05 mL, or faricimab 6.0 mg/0.05 mL [6,13,16]. After administering a topical anesthetic (0.4% oxybuprocaine hydrochloride; Benoxil™, Santen Pharmaceutical Co., Tokyo, Japan), the injections were performed using the standard pars plana approach (3.5 mm posterior to the limbus) with a 30-gauge needle under sterile conditions in a procedure room. 

### 2.3. Inclusion Criteria

The inclusion criteria were as follows: (1) a diagnosis of MNV due to nAMD; (2) recurrence of any fluid after IVA; (3) being switched to brolucizumab or faricimab due to one or more of the following reasons: insufficient improvement or maintenance of visual acuity, incomplete resolution of fluid, or difficulty in extending treatment intervals; and (4) completing three months of observation after switching. 

### 2.4. Exclusion Criteria

We referred to the exclusion criteria listed in previously published papers and adopted the following criteria for our study [21,22]: (1) history of treatments with photodynamic therapy, (2) any other retinal or optic nerve disease, and (3) presence of inflammatory or hereditary diseases that may induce MNV.

### 2.5. Outcomes

We measured the structural outcome (central macular thickness (CMT)) and the visual outcome (BCVA) at five different time points: just before (B0) and one month after the most recent IVA injection (B1), and just before (A0), one month after (A1) and three months after (A3) the first IVBr or IVF injection as shown in Figure 1. 

Based on the outcomes listed in previously reported papers, we used the following outcomes [18]:BCVA measured using the Landolt chart was converted into a logarithm of the minimum angle of resolution (logMAR) values for statistical analyses;CMT (μm) as measured with optical coherence tomography (OCT) (DRI OCT Triton, Topcon Inc., Tokyo, Japan) and defined as the distance from the internal limiting membrane to Bruch’s membrane at the fovea. For some cases who were not imaged by DRI OCT Triton, Cirrus high-definition-OCT (HD-OCT, Carl Zeiss Meditec Inc., Tokyo, Japan) was used.

### 2.6. Statistical Analyses

For statistical analyses, the baseline characteristics were compared using the unpaired *t*-test and Fisher’s exact tests. The Wilcoxon signed-rank test was used to determine the significance of the difference between the values before and after switching. The unpaired *t*-test was used to compare the changes in BCVAs and CMTs at each time point between the two groups. Descriptive statistics were used to describe the sample in terms of mean and standard deviation (SD). *p*  <  0.05 was considered statistically significant in all analyses. All analyses were performed using JMP Pro 17 software (SAS Institute, Cary, NC, USA).

## 3. Results

### 3.1. Patients’ Characteristics 

The baseline patient characteristics and clinical data are presented in Table 1. In the IVBr group, the mean age was 77.00 ± 5.81, and seven patients were women (35.00%). In the IVF group, the mean age was 74.87 ± 9.49, and five patients were women (33.33%). There were no statistically significant differences between the two groups for baseline data. The number of injections administered between the baseline switch (A0) and the three-month visit (A3) was 1.95 ± 0.78 in the IVBr group and 1.80 ± 0.77 in the IVF group (*p* = 0.3539). The presence of polypoidal choroidal vasculopathy (PCV) was determined based on indocyanine green angiography (ICGA). For cases not imaged by ICGA, the presence of PCV was determined based on OCT results according to previous consensus guidelines [23]. Subretinal fluid (SRF) on OCT at baseline was seen in 18 eyes (90.00%) in the IVBr group and in 11 (73.33%) in the IVF group (*p* = 0.3670). Intraretinal fluid (IRF) on OCT at baseline was seen in eight eyes (40.00%) in the IVBr group and in five eyes (33.33%) in the IVF group (*p* = 0.7372).

### 3.2. BCVA Outcomes

BCVA (logMAR) was 0.37 ± 0.35 at B0 and 0.38 ± 0.35 at B1 in the IVBr group and 0.31 ± 0.36 at B0 and 0.29 ± 0.37 at B1 in the IVF group, and the BCVA before switching treatments (between B0 and B1) showed no significant difference in both groups (*p* > 0.05 for all). BCVA showed more significant improvement at A1 (0.25 ± 0.34) and at A3 (0.19 ± 0.24) than at A0 (0.38 ± 0.35) (*p* = 0.0156, *p* = 0.0166, respectively) in the IVBr group. In the IVF group, BCVA at A1 (0.28 ± 0.42) and at A3 (0.33 ± 0.34) showed no significant improvement compared to at A0 (0.30 ± 0.36) (*p* > 0.05 for all) (Figure 2). There was no significant difference in BCVA improvement observed at either A1 or A3 between the two groups (*p* > 0.05 for all) (Figure 3).

### 3.3. CMT Outcomes

CMT (μm) was 305.85 ± 110.29 at B0 and 284.58± 75.11 at B1 in the IVBr group (*p* = 0.1034) and 264.15 ± 103.72 at B0 and 253.73 ± 64.00 at B1 in the IVF group (*p* = 0.3370). CMT showed more significant improvement at A1 (240.55 ± 51.82) and at A3 (243.21 ± 76.15) than at A0 (303.55 ± 79.18) (*p* = 0.0093, *p* = 0.0026, respectively) in the IVBr group. In the IVF group, CMT also showed more significant improvement at A1 (234.91 ± 47.29) and at A3 (250.50 ± 72.61) than at A0 (270.33 ± 77.62) (*p* = 0.0161, *p* = 0.0093, respectively) (Figure 4). There was no significant difference in CMT improvement observed at either A1 or A3 between the two groups (*p* > 0.05 for all) (Figure 5).

### 3.4. Representative Case

The following case study provides an example of the anatomical outcomes after switching to brolucizumab from aflibercept (Figure 6). A 78-year-old woman diagnosed with nAMD in 2020 required frequent intravitreal aflibercept injections, receiving a total of 10 injections. After the latest intravitreal aflibercept injection on 29 June 2023, she was switched to IVBr on 31 October 2023 due to insufficient resolution of pigment epithelial detachment (PED) (A0). The patient was monitored monthly for three months. One month after switching (A1), a significant reduction in PED height was observed, and the condition remained relatively stable even after three months (A3). The patient chose to continue IVBr treatment. On 18 April 2024, she received another IVBr treatment and maintained good disease control without any inflammation or complications.

### 3.5. Safety

No endophthalmitis, occlusive vasculitis, intraocular inflammation (IOI) or other ocular adverse events were seen, and no systemic adverse events were noted following the treatment switch.

## 4. Discussion

In the present study, we retrospectively evaluated the effects before and after switching to either brolucizumab or faricimab from aflibercept in eyes with nAMD in a clinical setting. Our study demonstrated that switching from aflibercept to either brolucizumab or faricimab improved structural outcomes in eyes with nAMD. No significant differences in visual and structural outcomes were observed between both groups throughout the follow-up period. To our knowledge, this report is the first to directly compare the outcomes of brolucizumab and faricimab in eyes with nAMD that have switched from aflibercept in routine clinical practice. 

In this study, although there were no significant differences, the IVBr group tended to show a greater reduction in CMT and greater improvement in BCVA compared to the IVF group. Similar to our study, Maruyama-Inoue et al. compared functional and morphological changes in the loading phase between patients with treatment-naïve nAMD treated with either brolucizumab or faricimab [24]. They showed brolucizumab had greater reductions in the central foveal thickness than faricimab, and brolucizumab also caused a trend toward faster visual improvements in the BCVA. Brolucizumab appears to be advantageous compared to faricimab in improving structural and visual outcomes in both treatment-naïve eyes and eyes switched from aflibercept. Despite structural improvement, no significant improvement in visual outcomes was reported when switched to faricimab in short-term follow-up, which was consistent with our study [21,25,26,27]. In these studies, photoreceptor damage at the time of switch was thought to be the reason for suboptimal improvement in BCVA. However, we found significant BCVA improvement when switched to brolucizumab. The rapid improvement in BCVA observed in the IVBr group may be attributed to the differences in mechanisms of action, molecular weight and VEGF affinity between the two agents, as mentioned in previous report [24]. Brolucizumab is a humanized single-chain antibody fragment that binds all the isoforms of VEGF-A, while faricimab is a bispecific molecule bound to an optimized Fc fragment that binds both VEGF-A and Ang-2 [28]. Brolucizumab, with a molecular weight of 26 kDa, is smaller than faricimab, which has a molecular weight of 146 kDa, potentially allowing for a higher drug concentration per injection [13,16]. This potentially enhances tissue penetration and increases its effectiveness compared to other agents. Additionally, the single-chain antibody fragment of brolucizumab enables its full binding capacity to VEGF. As a result, a higher number of molecules can be administered per injection within the same volume, leading to increased bioavailability. This may account for the greater reduction in CMT observed in the IVBr group compared to the IVF group. Faricimab helps restore vascular stability and promote vessel maturation by inhibiting Ang-2 [16,17]. Although a three-month period may have been insufficient for faricimab to fully demonstrate its potential, both treatments achieved significant reduction in CMT compared to baseline at 1 and 3 months.

Previous studies suggest that some patients require three monthly intravitreal injections to control retinal exudate, even after switching to a different anti-VEGF drug [20,25]. However, in our studies, the IVBr group received 1.95 ± 0.78 injections and the IVF group received 1.80 ± 0.77 injections within 3 months after the switch, which may not be sufficient. Further studies are needed to determine whether the loading dose regimen leads to further improvements in structural and visual outcomes in eyes with nAMD that have been switched from aflibercept.

IOI has become a clinical concern following the approval of brolucizumab for nAMD by the U.S. Food and Drug Administration in October 2019. The HAWK/HARRIER study reported that the incidence of brolucizumab-associated definite/probable IOI was 4.6% [29]. In this study, there were no complications such as IOI and RPE tears in both groups.

This study had several limitations. First, this study only evaluated a short period (3 months after the switch). Second, although no substantial differences were observed between the two groups in the number of injections within 3 months after the switch, the number of injections was determined by each physician and was not consistent. The timing of additional injections should be considered in future studies. Third, other small retrospective studies have reported that changing VEGF agents provided only a modest benefit, suggesting that switching VEGF agents may not necessarily be effective for the majority of patients [30]. Therefore, a large prospective study is needed to validate the present conclusions. Establishing a more optimal switching protocol could potentially lead to improved long-term visual outcomes and optimization of treatment for eyes with nAMD.

## 5. Conclusions

Switching from aflibercept to either brolucizumab or faricimab has a significant anatomical effect on eyes with nAMD and both treatments appear to be effective short-term treatment options. There is a trend towards greater visual improvements and reductions in CMT with brolucizumab. Further investigations may be needed with a larger sample size, longer observation period, and standardized injection regimen to determine the optimal switching treatment option from aflibercept in clinical practice.

## Figures and Tables

**Figure 1 medicina-60-01170-f001:**
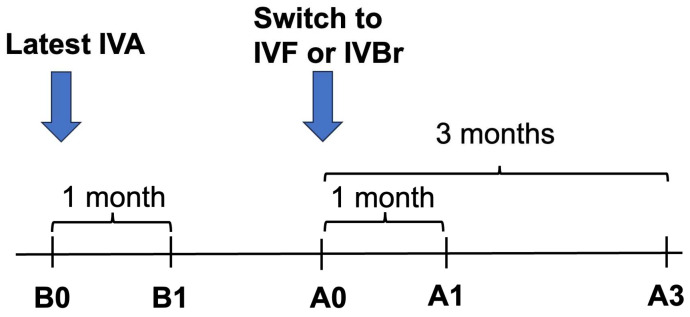
Timepoints evaluated in this study. Patients were evaluated at five time points: just before (B0) and one month after the most recent intravitreal aflibercept (IVA) injection (B1), and just before (A0), one month after (A1) and three months after (A3) the first intravitreal brolucizumab (IVBr) or intravitreal faricimab (IVF) injection.

**Figure 2 medicina-60-01170-f002:**
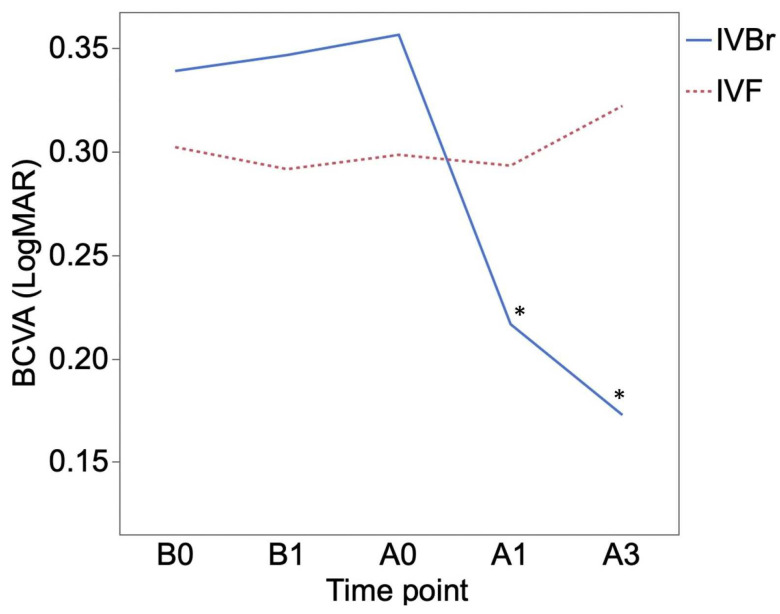
The mean best-corrected visual acuity (BCVA) of the IVBr and IVF groups at each time point. Both groups showed no significant improvement in BCVA at B1 compared to at B0 (*p* > 0.05 for all). The IVBr group showed significant improvements in BCVA at A1 and at A3 compared to at A0 (A1, *p* = 0.0156; A3, *p* = 0.0166). No significant improvement in BCVA was observed in the IVF group at A1 and at A3 compared to at A0 (*p* > 0.05 for all). *: *p* < 0.05.

**Figure 3 medicina-60-01170-f003:**
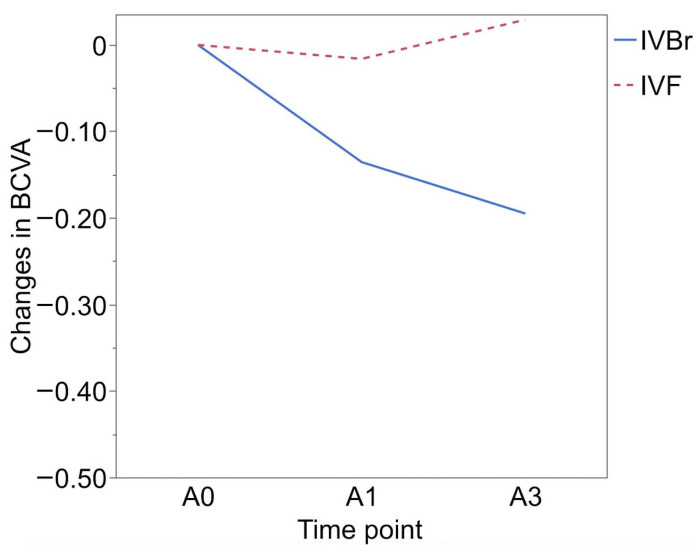
There was no significant difference in BCVA improvement observed at either A1 or A3 between two groups (*p* > 0.05 for all).

**Figure 4 medicina-60-01170-f004:**
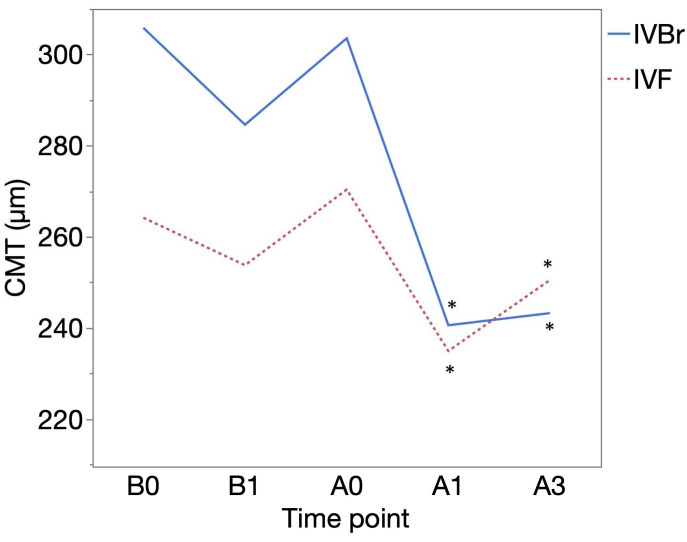
The mean central macular thickness (CMT (μm)) of the IVBr and IVF groups at each time point. No significant improvements in CMT were observed at B1 compared to B0 in either group (*p* > 0.05 for all). Both groups showed significant improvements in CMT at A1 (IVF, *p* = 0.0093; IVBr, *p* = 0.0161) and at A3 (IVF, *p* = 0.0026; IVBr, *p* = 0.0093) compared to at A0. *: *p* < 0.05.

**Figure 5 medicina-60-01170-f005:**
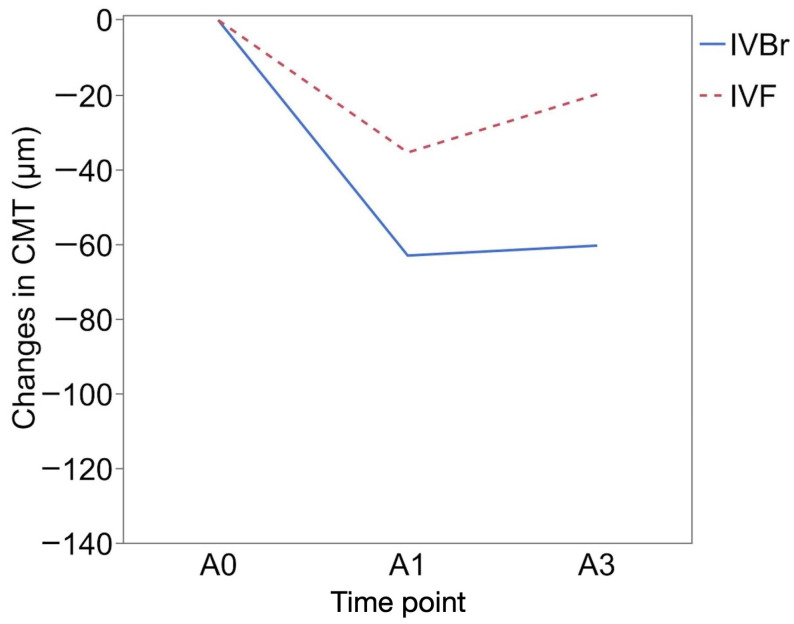
There was no significant difference in CMT improvement observed at either A1 or A3 between the two groups (*p* > 0.05 for all).

**Figure 6 medicina-60-01170-f006:**
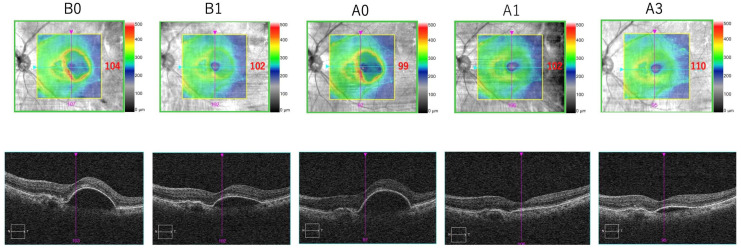
A representative case of switching from IVA to IVBr in eyes with nAMD. Optical coherence tomographic scans going through the fovea at **B0**, **B1**, **A0**, **A1** and **A3**. After switching to brolucizumab from aflibercept, decreases in CMT and the height of the pigment epithelial detachment were observed. N, nasal; T, temporal.

**Table 1 medicina-60-01170-t001:** Baseline (just before switching) characteristics of patients or eyes immediately before switching.

Characteristics		IVBr-Injected Group (*n* = 20)	IVF-Injected Group (*n* = 15)	*p* Value
Age, years	Mean ± SD	77.00 ± 5.81	74.87 ± 9.49	0.5125
Sex	Female, *n* (%)	7 (35.00%)	5 (33.33%)	1.0000
Lens	pseudophakia, *n* (%)	15 (75.00%)	7 (46.67%)	0.1567
Previous history of PPV, eyes	*n* (%)	3 (15.00%)	2 (13.33%)	1.0000
PCV, eyes	*n* (%)	2 (10.00%)	0 (0%)	0.4958
SRF, eyes	*n* (%)	18 (90.00%)	11 (73.33%)	0.3670
IRF, eyes	*n* (%)	8 (40.00%)	5 (33.33%)	0.7372
CMT, μm	Mean ± SD	305.85 ± 110.29	270.33 ± 77.62	0.2243
BCVA (logMAR)	Mean ± SD	0.38 ± 0.35	0.28 ± 0.42	0.4988
Number of prior IVA doses before switching	Mean ± SD	7.35 ± 4.64	5.87 ± 3.50	0.7508
Period from the last IVA, weeks	Mean ± SD	9.78 ± 3.76	9.65 ± 4.83	0.6866

BCVA, best-corrected visual acuity; CMT, central macular thickness; IRF, intraretinal fluid; IVA, intravitreal aflibercept; IVBr, intravitreal brolucizumab; IVF, intravitreal faricimab; logMAR, logarithm of the minimum angle of resolution; PCV, polypoidal choroidal vasculopathy; PPV, pars plana vitrectomy; SD, standard deviation; SRF, subretinal fluid.

## Data Availability

The data used and analyzed for this study are available from the corresponding author on reasonable request.

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
