# Peer review of "Short-Term Comparison of Switching to Brolucizumab or Faricimab from Aflibercept in Neovascular AMD Patients"

_medicina, 2024, doi:10.3390/medicina60071170_

Round 1

Reviewer 1 Report

Comments and Suggestions for Authors

Many thanks to the opportunity to review the paper of Dr Kin. In their study, authors report short-term outcomes of switching to brolucizumab and faricimab from aflibercept. I found the paper interesting and practically useful. However, some modifications are needed before further consideration of the manuscript.

1.      Term “resistant” is not fully applicable since study eyes had quite long period after last aflibercept so it is not surprising to see some activity. I agree that all we seeking for better outcomes and longer treatment intervals but technically it is not “resistance”, which may include situation where dry macula is not achieved in bimonthly injections or more frequently.

2.      Statistics for baseline types of fluids (before switching) is desirable

3.      Figure 1 is very important but somewhat misleading since shows 16 weeks interval before switching while it was app 10 in fact. This may lead to the wrong conclusion that pre-switch time was longer than post-switch what makes results incompatible.

4.      Table 1: parameter “RPE disturbance” looks strange. No definition, no other mentioning…

5.      How diagnosis of PCV was established?

6.      Description of IVI technique is redundant 

Author Response

Firstly, we would like to thank all the reviewers and a handling-editor for their carefully reading our manuscript and constructive feedback. We found their comments very insightful and valuable. In light of these comments, we have improved our manuscript. We are confident that with these revisions, we have now satisfied most reviewer comments. With these changes, we humbly hope the reviewers and handling-editor will agree that our revised manuscript is now suitable for publication.

Please find below our point-by-point response to your comments for your review and consideration.

  1. Term “resistant” is not fully applicable since study eyes had quite long period after last aflibercept so it is not surprising to see some activity. I agree that all we seeking for better outcomes and longer treatment intervals but technically it is not “resistance”, which may include situation where dry macula is not achieved in bimonthly injections or more frequently.

Response

Thank you for your insightful feedback on our manuscript. After careful consideration, we agree that the term "resistant" is not fully applicable to our study population. Therefore, we have revised the manuscript to reflect that our study focuses on cases where patients were switched from aflibercept to brolucizumab or faricimab.

We are grateful for your important comment, which has significantly improved the accuracy and clarity of our work.

Thank you so much for your valuable input.

  1. Statistics for baseline types of fluids (before switching) is desirable.

Response

Thank you for this valuable comment. As you pointed out, we have calculated statistics for baseline types of fluids (before switching) and we have added to Table1 (Line 152) and Section 3.1 (Line 142-144) in the revised manuscript.

Thank you for helping us improve the clarity and accuracy of our work.

  1. Figure 1 is very important but somewhat misleading since shows 16 weeks interval before switching while it was app 10 in fact. This may lead to the wrong conclusion that pre-switch time was longer than post-switch what makes results incompatible.

Response

Thank you for your insightful feedback. We understand your concern about the potentially misleading presentation of the interval before switching treatments. The additional post-switch injections were administered based on the discretion of the treating physician rather than solely on treatment efficacy. Therefore, the difference in injection intervals before and after the switch cannot be directly compared in this study.

We have acknowledged this limitation and added a sentence emphasizing that the timing of additional injections should be considered in future studies in the revised manuscript (Line 272-273). We hope this clarifies the context and rationale behind our approach.

Thank you once again for your valuable input, which has helped us refine our work.

  1. Table 1: parameter “RPE disturbance” looks strange. No definition, no other mentioning…

Response

Thank you for your valuable feedback on our manuscript. Upon careful consideration, we have decided to remove this parameter from the table as it does not directly relate to the results presented in our study.

Thank you for helping us improve the clarity and accuracy of our work.

  1. How diagnosis of PCV was established?

Response

Thank you for your valuable feedback on our manuscript. We determined the presence of polypoidal choroidal vasculopathy (PCV) based on indocyanine green angiography (ICGA). For cases not imaged by ICGA, the presence of PCV was determined based on optical coherence tomography (OCT) results according to previous consensus guidelines (Cheung et al., Ophthalmology, 2020). We have added this statement to Section 3.1 (Line 139-142).

Thank you once again for your insightful comments.

  1. Description of IVI technique is redundant. 

Response

Thank you for your insightful feedback on our manuscript. We acknowledge that the description of the intravitreal injection technique was redundant. We have revised the manuscript to address this issue and streamlined the relevant section accordingly (Line 87-91).

We appreciate your guidance in helping us improve the clarity and conciseness of our work.

Reviewer 2 Report

Comments and Suggestions for Authors

The Article “Short-term Comparison of Switch to Brolucizumab or Faricimab in Eyes with Aflibercept-Resistant Neovascular AMD” focuses to evaluate observational clinical study included 20 eyes of 20 patients switched to brolucizumab and 15 eyes of 14 patients switched to faricimab for aflibercept-resistant age-related macular degeneration and compare the changes in visual and structural results in patients with intravitreal aflibercept (IVA)-resistant neovascular age-related macular degeneration switched to either intravitreal brolucizumab or intravitreal faricimab injections in a clinical setting. The work represented with appropriate section like introduction, material methods, result, discussion and conclusion. The content is well in scope of journal and contains finding details in tabular and figures. The results discussed with the available literature too.

The Comments are as follows to improve the manuscript:

1.      Introduction, please add some state of number of patients suffer from Resistant Neovascular AMD disease.

2.      Introduction is very limited, please expand with the current therapy, limitations and various combinations used in treatment of Resistant Neovascular AMD.

3.      Line 58: [12-20], here bulk citation is there, keep only most relevant and recent one. Avoid bulk citation.

4.      Line 76-77: Aflibercept 2.0 mg/0.05 mL, brolucizumab 6.0 mg/0.05 mL or faricimab 6.0 mg/0.05 76 mL was injected into the vitreous cavity of all patients. Please cite the text for dose selection.

5.      Line 82: Following the treatment, patients were prescribed 0.5% MFLX ophthalmic solutions. What is MFLX here?

6.      Please cite the content of inclusion and exclusion criteria if supported from previous literature.

7.      Line 102: Outcome measures included….. Please cite the content of outcome measures if it’s from previous study.

8.      Figure 2,3,4 and 5 need to improve resolution and increase in font in the figure to improve readability.

9.      Very limited, discuss for 3.4. Representative Case, figure 6. Please improve that.

Author Response

Firstly, we would like to thank all the reviewers and a handling-editor for their carefully reading our manuscript and constructive feedback. We found their comments very insightful and valuable. In light of these comments, we have improved our manuscript. We are confident that with these revisions, we have now satisfied most reviewer comments. With these changes, we humbly hope the reviewers and handling-editor will agree that our revised manuscript is now suitable for publication.

Please find below our point-by-point response to your comments for your review and consideration.

  1. Introduction, please add some state of number of patients suffer from Resistant Neovascular AMD disease.

Response

Thank you for your valuable feedback on our manuscript. We appreciate your important comment regarding the need to include data on patients suffering from Resistant Neovascular AMD disease. In response, we have added this information to the Introduction section. Additionally, we have included details about tachyphylaxis to provide a more comprehensive context (Line 42-48).

Thank you for your insightful comments, which have significantly enhanced the depth and relevance of our work.

  1. Introduction is very limited, please expand with the current therapy, limitations and various combinations used in treatment of Resistant Neovascular AMD.

Response

Thank you for your valuable feedback on our manuscript. Following your advice, we have expanded the Introduction to include information on current therapies, their limitations, and various combinations used in the treatment of Resistant Neovascular AMD such as switching between anti-VEGF agents, adjusting treatment regimens, exploring combination therapies, and employing high-dose treatments (Line 49-54). This addition has significantly enriched the Introduction section, and we are very grateful for your insightful suggestions.

Thank you once again for your guidance in improving our manuscript.

  1. Line 58: [12-20], here bulk citation is there, keep only most relevant and recent one. Avoid bulk citation.

Response

Thank you for your insightful feedback regarding the citations in our manuscript. We appreciate you pointing out that the inclusion of eight references was excessive. We have now refined the citations and included only the three most relevant and recent references (Line 68).

Thank you for your guidance in helping us improve the clarity and precision of our manuscript.

  1. Line 76-77: Aflibercept 2.0 mg/0.05 mL, brolucizumab 6.0 mg/0.05 mL or faricimab 6.0 mg/0.05 76 mL was injected into the vitreous cavity of all patients. Please cite the text for dose selection.

Response

Thank you for your valuable feedback on our manuscript. We appreciate your observation regarding the need for a citation for the dose selection of aflibercept, brolucizumab, and faricimab. We have now included the appropriate references to support the dosage used in our study (Line 87-88).

Thank you for helping us improve the accuracy and completeness of our manuscript.

  1. Line 82: Following the treatment, patients were prescribed 0.5% MFLX ophthalmic solutions. What is MFLX here?

Response

Thank you for your valuable feedback on our manuscript. We appreciate you pointing out that the abbreviation "MFLX" was not defined. As the description of intravitreal injection technique was redundant, we have deleted from the main text (Line 87-91).

Thank you for helping us improve the clarity and accuracy of our work.

  1. Please cite the content of inclusion and exclusion criteria if supported from previous literature.

Response

Thank you for your valuable feedback on our manuscript. In response to your comment, we have added citations to support the content of the exclusion criteria, as they were based on previous literature (Line 99-100).

Thank you for helping us enhance the rigor and credibility of our work.

  1. Line 102: Outcome measures included….. Please cite the content of outcome measures if it’s from previous study

Response

Thank you for your valuable feedback on our manuscript. We appreciate you pointing out that the citation for the content of outcome measures were needed if it’s from previous study. We have now included the appropriate references in the revised manuscript (Line 113-114).

Thank you for helping us improve the clarity and accuracy of our work.

  1. Figure 2,3,4 and 5 need to improve resolution and increase in font in the figure to improve readability.

Response

Thank you for your valuable feedback on our manuscript. We appreciate you pointing out that Figure 2, 3, 4 and 5 need to improve resolution and increase in font in the figures to improve readability. We have now improved resolution and increase in font in the figures in the revised manuscript (Line 172,178,189,194).

We appreciate your guidance in helping us improve the clarity and conciseness of our work.

  1. Very limited, discuss for 3.4. Representative Case, figure 6. Please improve that.

Response

Thank you for your valuable feedback on our manuscript. We appreciate your comment regarding the limited discussion of the Representative Case, Figure 6. In response, we have expanded this section by adding detailed information about the treatment course and outcomes in a case report format (Line 198-207).

Thank you for helping us enhance the comprehensiveness and depth of our manuscript.